# Deep Neural Networks for Object Detection

**Christian Szegedy    Alexander Toshev    Dumitru Erhan**
Google, Inc.
{szegedy, toshev, dumitru}@google.com

## Abstract

Deep Neural Networks (DNNs) have recently shown outstanding performance on image classification tasks [14]. In this paper we go one step further and address the problem of object detection using DNNs, that is not only classifying but also precisely localizing objects of various classes. We present a simple and yet powerful formulation of object detection as a regression problem to object bounding box masks. We define a multi-scale inference procedure which is able to produce high-resolution object detections at a low cost by a few network applications. State-of-the-art performance of the approach is shown on Pascal VOC.

## 1   Introduction

As we move towards more complete image understanding, having more precise and detailed object recognition becomes crucial. In this context, one cares not only about classifying images, but also about precisely estimating estimating the class and location of objects contained within the images, a problem known as object detection.

The main advances in object detection were achieved thanks to improvements in object representations and machine learning models. A prominent example of a state-of-the-art detection system is the Deformable Part-based Model (DPM) [9]. It builds on carefully designed representations and kinematically inspired part decompositions of objects, expressed as a graphical model. Using discriminative learning of graphical models allows for building high-precision part-based models for variety of object classes.

Manually engineered representations in conjunction with shallow discriminatively trained models have been among the best performing paradigms for the related problem of object classification as well [17]. In the last years, however, Deep Neural Networks (DNNs) [12] have emerged as a powerful machine learning model.

DNNs exhibit major differences from traditional approaches for classification. First, they are deep architectures which have the capacity to learn more complex models than shallow ones [2]. This expressivity and robust training algorithms allow for learning powerful object representations without the need to hand design features. This has been empirically demonstrated on the challenging ImageNet classification task [5] across thousands of classes [14, 15].

In this paper, we exploit the power of DNNs for the problem of object detection, where we not only classify but also try to *precisely localize objects*. The problem we are address here is challenging, since we want to detect a *potentially large number object instances with varying sizes in the same image* using a limited amount of computing resources.

We present a formulation which is capable of predicting the bounding boxes of multiple objects in a given image. More precisely, we formulate a DNN-based regression which outputs a binary mask of the object bounding box (and portions of the box as well), as shown in Fig. 1. Additionally, we employ a simple bounding box inference to extract detections from the masks. To increase localization precision, we apply the DNN mask generation in a multi-scale fashion on the full image as well as on a small number of large image crops, followed by a refinement step (see Fig. 2).

In this way, only through a few dozen DNN-regressions we can achieve state-of-art bounding box localization.

In this paper, we demonstrate that DNN-based regression is capable of learning features which are not only good for classification, but also capture *strong geometric information*. We use the general architecture introduced for classification by [14] and replace the last layer with a regression layer. The somewhat surprising but powerful insight is that networks which to some extent encode translation invariance, can capture object locations as well.

Second, we introduce a multi-scale box inference followed by a refinement step to produce precise detections. In this way, we are able to apply a DNN which predicts a low-resolution mask, limited by the output layer size, to pixel-wise precision at a low cost – the network is a applied only a few dozen times per input image.

In addition, the presented method is quite simple. There is no need to hand design a model which captures parts and their relations explicitly. This simplicity has the advantage of easy applicability to wide range of classes, but also show better detection performance across a wider range of objects – rigid ones as well as deformable ones. This is presented together with state-of-the-art detection results on Pascal VOC challenge [7] in Sec. 7.

## 2   Related Work

One of the most heavily studied paradigms for object detection is the deformable part-based model, with [9] being the most prominent example. This method combines a set of discriminatively trained parts in a star model called pictorial structure. It can be considered as a 2-layer model – parts being the first layer and the star model being the second layer. Contrary to DNNs, whose layers are generic, the work by [9] exploits domain knowledge – the parts are based on manually designed Histogram of Gradients (HOG) descriptors [4] and the structure of the parts is kinematically motivated.

Deep architectures for object detection and parsing have been motivated by part-based models and traditionally are called compositional models, where the object is expressed as layered composition of image primitives. A notable example is the $And/Or$ graph [20], where an object is modeled by a tree with $And$-nodes representing different parts and $Or$-nodes representing different modes of the same part. Similarly to DNNs, the $And/Or$ graph consists of multiple layers, where lower layers represent small generic image primitives, while higher layers represent object parts. Such compositional models are easier to interpret than DNNs. On the other hand, they require inference while the DNN models considered in this paper are purely feed-forward with no latent variables to be inferred.

Further examples of compositional models for detection are based on segments as primitives [1], focus on shape [13], use Gabor filters [10] or larger HOG filters [19]. These approaches are traditionally challenged by the difficulty of training and use specially designed learning procedures. Moreover, at inference time they combine bottom-up and top-down processes.

Neural networks (NNs) can be considered as compositional models where the nodes are more generic and less interpretable than the above models. Applications of NNs to vision problems are decades old, with Convolutional NNs being the most prominent example [16]. It was not until recently than these models emerged as highly successful on large-scale image classification tasks [14, 15] in the form of DNNs. Their application to detection, however, is limited. Scene parsing, as a more detailed form of detection, has been attempted using multi-layer Convolutional NNs [8]. Segmentation of medical imagery has been addressed using DNNs [3]. Both approaches, however, use the NNs as local or semi-local classifiers either over superpixels or at each pixel location. Our approach, however, uses the full image as an input and performs localization through regression. As such, it is a more efficient application of NNs.

Perhaps the closest approach to ours is [18] which has similar high level objective but use much smaller network with a different features, loss function and without a machinery to distinguish between multiple instances of the same class.

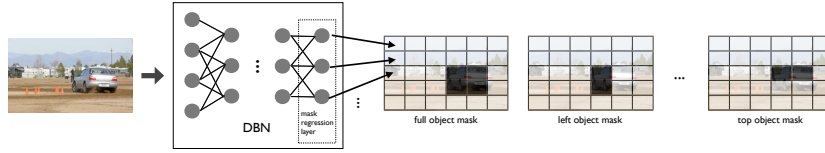

Figure 1: A schematic view of object detection as DNN-based regression.

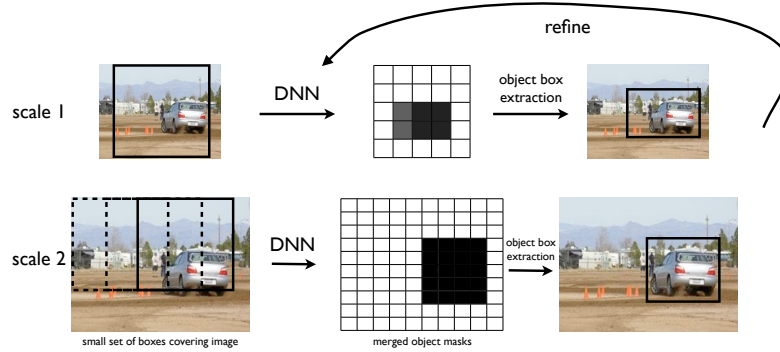

Figure 2: After regressing to object masks across several scales and large image boxes, we perform object box extraction. The obtained boxes are refined by repeating the same procedure on the sub images, cropped via the current object boxes. For brevity, we display only the full object mask, however, we use all five object masks.

## 3   DNN-based Detection

The core of our approach is a DNN-based regression towards an object mask, as shown in Fig. 1. Based on this regression model, we can generate masks for the full object as well as portions of the object. A single DNN regression can give us masks of multiple objects in an image. To further increase the precision of the localization, we apply the DNN localizer on a small set of large sub-windows. The full flow is presented in Fig. 2 and explained below.

## 4   Detection as DNN Regression

Our network is based on the convolutional DNN defined by [14]. It consists of total 7 layers, the first 5 of which being convolutional and the last 2 fully connected. Each layer uses a rectified linear unit as a non-linear transformation. Three of the convolutional layers have in addition max pooling. For further details, we refer the reader to [14].

We adapt the above generic architecture for localization. Instead of using a softmax classifier as a last layer, we use a regression layer which generates an object binary mask $DNN(x; \Theta) \in \mathbb{R}^N$, where $\Theta$ are the parameters of the network and $N$ is the total number of pixels. Since the output of the network has a fixed dimension, we predict a mask of a fixed size $N = d \times d$. After being resized to the image size, the resulting binary mask represents one or several objects: it should have value 1 at particular pixel if this pixel lies within the bounding box of an object of a given class and 0 otherwise.

The network is trained by minimizing the $L_2$ error for predicting a ground truth mask $m \in [0, 1]^N$ for an image $x$:

$$\min_{\Theta} \sum_{(x,m) \in D} ||(Diag(m) + \lambda I)^{1/2}(DNN(x; \Theta) - m)||_2^2,$$

where the sum ranges over a training set $D$ of images containing bounding boxed objects which are represented as binary masks.

Since our base network is highly non-convex and optimality cannot be guaranteed, it is sometimes necessary to regularize the loss function by using varying weights for each output depending on the

ground truth mask. The intuition is that most of the objects are small relative to the image size and the network can be easily trapped by the trivial solution of assigning a zero value to every output. To avoid this undesirable behavior, it is helpful to increase the weight of the outputs corresponding to non-zero values in the ground truth mask by a parameter $\lambda \in \mathbb{R}^+$. If $\lambda$ is chosen small, then the errors on the output with groundtruth value $0$ are penalized significantly less than those with $1$ and therefore encouraging the network to predict nonzero values even if the signals are weak.

In our implementation, we used networks with a receptive field of $225 \times 225$ and outputs predicting a mask of size $d \times d$ for $d = 24$.

# 5 Precise Object Localization via DNN-generated Masks

Although the presented approach is capable of generating high-quality masks, there are several additional challenges. First, a single object mask might not be sufficient to disambiguate objects which are placed next to each other. Second, due to the limits in the output size, we generate masks that are much smaller than the size of the original image. For example, for an image of size $400 \times 400$ and $d = 24$, each output would correspond to a cell of size $16 \times 16$ which would be insufficient to precisely localize an object, especially if it is a small one. Finally, since we use as an input the full image, small objects will affect very few input neurons and thus will be hard to recognize. In the following, we explain how we address these issues.

## 5.1 Multiple Masks for Robust Localization

To deal with multiple touching objects, we generate not one but several masks, each representing either the full object or part of it. Since our end goal is to produce a bounding box, we use one network to predict the object box mask and four additional networks to predict four halves of the box: bottom, top, left and right halves, all denoted by $m^h$, $h \in \{\text{full, bottom, top, left, left}\}$. These five predictions are over-complete but help reduce uncertainty and deal with mistakes in some of the masks. Further, if two objects of the same type are placed next to each other, then at least two of the produced five masks would not have the objects merged which would allow to disambiguate them. This would enable the detection of multiple objects.

At training time, we need to convert the object box to these five masks. Since the masks can be much smaller than the original image, we need to downsize the ground truth mask to the size of the network output. Denote by $T(i, j)$ the rectangle in the image for which the presence of an object is predicted by output $(i, j)$ of the network. This rectangle has upper left corner at $\left( \frac{d_1}{d}(i-1), \frac{d_2}{d}(j-1) \right)$ and has size $\frac{d_1}{d} \times \frac{d_1}{d}$, where $d$ is the size of the output mask and $d_1, d_2$ the height and width of the image. During training we assign as value $m(i, j)$ to be predicted as portion of $T(i, j)$ being covered by box $bb(h)$ :

$$m^h(i, j; bb) = \frac{area(bb(h) \cap T(i, j))}{area(T(i, j))} \tag{1}$$

where $bb(\text{full})$ corresponds to the ground truth object box. For the remaining values of $h$, $bb(h)$ corresponds to the four halves of the original box.

Note that we use the full object box as well as the top, bottom, left and right halves of the box to define total five different coverage types. The resulting $m^h(bb)$ for groundtruth box $bb$ are being used at training time for network of type $h$.

At this point, it should be noted that one could train one network for all masks where the output layer would generate all five of them. This would enable scalability. In this way, the five localizers would share most of the layers and thus would share features, which seems natural since they are dealing with the same object. An even more aggressive approach — using the same localizer for a lot of distinct classes – seems also workable.

## 5.2 Object Localization from DNN Output

In order to complete the detection process, we need to estimate a set of bounding boxes for each image. Although the output resolution is smaller than the input image, we rescale the binary masks to the resolution as the input image. The goal is to estimate bounding boxes $bb = (i, j, k, l)$ parametrized by their upper-left corner $(i, j)$ and lower-right corner $(k, l)$ in output mask coordinates.

To do this, we use a score $S$ expressing an agreement of each bounding box $bb$ with the masks and infer the boxes with highest scores. A natural agreement would be to measure what portion of the bounding box is covered by the mask:

$$S(bb, m) = \frac{1}{area(bb)} \sum_{(i,j)} m(i,j) area(bb \cap T(i,j)) \tag{2}$$

where we sum over all network outputs indexed by $(i, j)$ and denote by $m = DNN(x)$ the output of the network. If we expand the above score over all five mask types, then final score reads:

$$S(bb) = \sum_{h \in halves} (S(bb(h), m^h) - S(bb(\bar{h}), m^h)) \tag{3}$$

where $halves = \{\text{full}, \text{bottom}, \text{top}, \text{left}, \text{left}\}$ index the full box and its four halves. For $h$ denoting one of the halves $\bar{h}$ denotes the opposite half of $h$, e.g. a top mask should be well covered by a top mask and not at all by the bottom one. For $h = \text{full}$, we denote by $\bar{h}$ a rectangular region around $bb$ whose score will penalize if the full masks extend outside $bb$. In the above summation, the score for a box would be large if it is consistent with all five masks.

We use the score from Eq. (3) to exhaustively search in the set of possible bounding boxes. We consider bounding boxes with mean dimension equal to $[0.1, \ldots, 0.9]$ of the mean image dimension and 10 different aspect ratios estimated by k-means clustering of the boxes of the objects in the training data. We slide each of the above 90 boxes using stride of 5 pixels in the image. Note that the score from Eq. (3) can be efficiently computed using 4 operations after the integral image of the mask $m$ has been computed. The exact number of operations is $5(2 \times \#\text{pixels} + 20 \times \#\text{boxes})$, where the first term measures the complexity of the integral mask computation while the second accounts for box score computation.

To produce the final set of detections we perform two types of filtering. The first is by keeping boxes with strong score as defined by Eq. (2), e.g. larger than $0.5$. We further prune them by applying a DNN classifier by [14] trained on the classes of interest and retaining the positively classified ones w.r.t to the class of the current detector. Finally, we apply non-maximum suppression as in [9].

### 5.3 Multi-scale Refinement of DNN Localizer

The issue with insufficient resolution of the network output is addressed in two ways: (i) applying the DNN localizer over several scales and a few large sub-windows; (ii) refinement of detections by applying the DNN localizer on the top inferred bounding boxes (see Fig. 2).

Using large windows at various scales, we produce several masks and merge them into higher resolution masks, one for each scale. The range of the suitable scales depends on the resolution of the image and the size of the receptive field of the localizer - we want the image be covered by network outputs which operate at a higher resolution, while at the same time we want each object to fall within at least one window and the number of these windows to be small.

To achieve the above goals, we use three scales: the full image and two other scales such that the size of the window at a given scale is half of the size of the window at the previous scale. We cover the image at each scale with windows such that these windows have a small overlap – 20% of their area. These windows are relatively small in number and cover the image at several scales. Most importantly, the windows at the smallest scale allow localization at a higher resolution.

At inference time, we apply the DNN on all windows. Note that it is quite different from sliding window approaches because we need to evaluate a small number of windows per image, usually less than $40$. The generated object masks at each scale are merged by maximum operation. This gives us three masks of the size of the image, each 'looking' at objects of different sizes. For each scale, we apply the bounding box inference from Sec. 5.2 to arrive at a set of detections. In our implementation, we took the top 5 detections per scale, resulting in a total of 15 detections.

To further improve the localization, we go through a second stage of DNN regression called refinement. The DNN localizer is applied on the windows defined by the initial detection stage – each of the 15 bounding boxes is enlarged by a factor of 1.2 and is applied to the network. Applying the localizer at higher resolution increases the precision of the detections significantly.

The complete algorithm is outlined in Algorithm 1.

---

**Algorithm 1:** Overall algorithm: multi-scale DNN-based localization and subsequent refinement. The above algorithm is applied for each object class separately.

---

**Input**: $x$ input image of size; networks $DNN^h$ producing full and partial object box mask.
**Output**: Set of detected object bounding boxes with confidence scores.
$detections \leftarrow \emptyset$
$scales \leftarrow$ compute suitable scales for image.
**for** $s \in scales$ **do**
    $windows \leftarrow$ generate windows for the given scale $s$.
    **for** $w \in windows$ **do**
        **for** $h \in \{lower, upper, top, bottom, full\}$ **do**
            $m_w^h \leftarrow DNN^h(w)$
        **end**
    **end**
    $m^h \leftarrow$ merge masks $m_w^h$, $w \in windows$
    $detections_s \leftarrow$ obtain a set of bounding boxes with scores from $m^h$ as in Sec. 5.2
    $detections \leftarrow detections \cup detections_s$
**end**
$refined \leftarrow \emptyset$
**for** $d \leftarrow detections$ **do**
    $c \leftarrow$ cropped image for enlarged bounding box of $d$
    **for** $h \in \{lower, upper, top, bottom, full\}$ **do**
        $m_w^h \leftarrow DNN^h(c)$
    **end**
    $\overline{detection} \leftarrow$ infer highest scoring bounding box from $m^h$ as in Sec. 5.2
    $refined \leftarrow refined \cup \overline{detection}$
**end**
**return** $refined$

---

## 6   DNN Training

One of the compelling features of our network is its simplicity: the classifier is simply replaced by a mask generation layer without any smoothness prior or convolutional structure. However, it needs to be trained with a huge amount of training data: objects of different sizes need to occur at almost every location.

For training the *mask generator*, we generate several thousand samples from each image divided into 60% negative and 40% positive samples. A sample is considered to be negative if it does not intersect the bounding box of any object of interest. Positive samples are those covering at least 80% of the area of some of the object bounding boxes. The crops are sampled such that their width is distributed uniformly between the prescribed minimum scale and the width of the whole image.

We use similar preparations steps to train the *classifier* used for the final pruning of our detections. Again, we sample several thousand samples from each image: 60% negative and 40% positive samples. The negative samples are those whose bounding boxes have less than 0.2 Jaccard-similarity with any of the groundtruth object boxes The positive samples must have at least 0.6 similarity with some of the object bounding boxes and are labeled by the class of the object with most similar bounding box to the crop. Adding the extra negative class acts as a regularizer and improves the quality of the filters. In both cases, the total number of samples is chosen to be ten million for each class.

Since training for localization is harder than classification, it is important to start with the weights of a model with high quality low-level filters. To achieve this, we first train the network for classification and reuse the weights of all layers but the classifier for localization. For localization, we we have fine-tuned the whole network, including the convolutional layers.

The networks were trained by stochastic gradient using ADAGRAD [6] to estimate the learning rate of the layers automatically.

| class | aero | bicycle | bird | boat | bottle | bus | car | cat | chair | cow |
|---|---|---|---|---|---|---|---|---|---|---|
| DetectorNet[1] | .292 | .352 | **.194** | .167 | .037 | **.532** | .502 | **.272** | .102 | **.348** |
| Sliding windows[1] | .213 | .190 | .068 | .120 | .058 | .294 | .237 | .101 | .059 | .131 |
| 3-layer model [19] | .294 | .558 | .094 | .143 | **.286** | .440 | .513 | .213 | .200 | .193 |
| Felz. et al. [9] | **.328** | .568 | .025 | **.168** | .285 | .397 | .516 | .213 | .179 | .185 |
| Girshick et al. [11] | .324 | **.577** | .107 | .157 | .253 | .513 | **.542** | .179 | **.210** | .240 |
| class | table | dog | horse | m-bike | person | plant | sheep | sofa | train | tv |
| DetectorNet[1] | **.302** | **.282** | .466 | .417 | .262 | .103 | **.328** | .268 | .398 | **.470** |
| Sliding windows[1] | .110 | .134 | .220 | .243 | .173 | .070 | .118 | .166 | .240 | .119 |
| 3-layer model [19] | .252 | .125 | .504 | .384 | .366 | **.151** | .197 | .251 | .368 | .393 |
| Felz. et al. [9] | .259 | .088 | .492 | .412 | .368 | .146 | .162 | .244 | .392 | .391 |
| Girshick et al. [11] | .257 | .116 | **.556** | **.475** | **.435** | .145 | .226 | **.342** | **.442** | .413 |

Table 1: Average precision on Pascal VOC2007 test set.

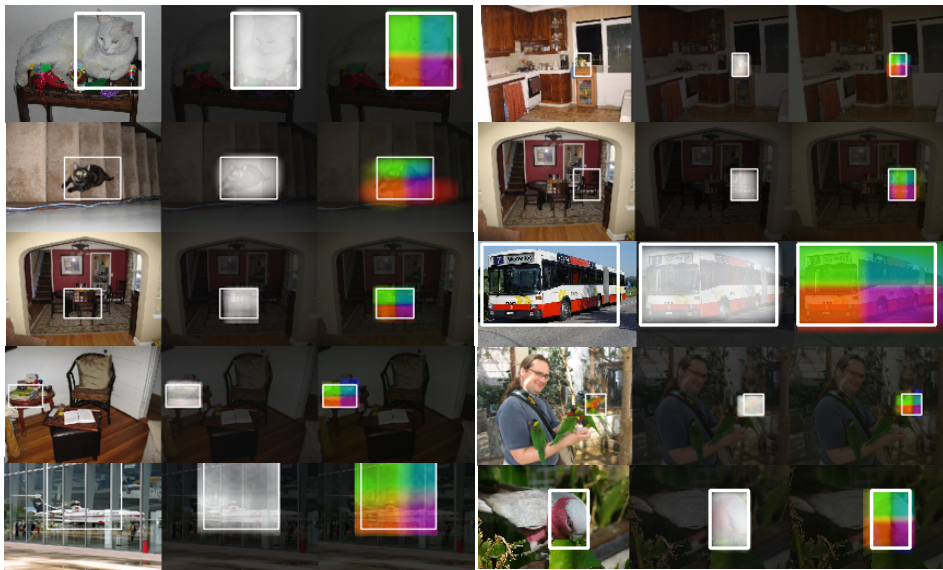

Figure 3: For each image, we show two heat maps on the right: the first one corresponds to the output of $DNN^{\text{full}}$, while the second one encodes the four partial masks in terms of the strength of the colors red, green, blue and yellow. In addition, we visualize the estimated object bounding box. All examples are correct detections with exception of the examples in the last row.

## 7 Experiments

**Dataset:** We evaluate the performance of the proposed approach on the test set of the Pascal Visual Object Challenge (VOC) 2007 [7]. The dataset contains approx. 5000 test images over 20 classes. Since our approach has large number of parameters, we train on the VOC2012 training and validation set which has approx. 11K images. At test time an algorithm produces for an image a set of detections, defined bounding boxes and their class labels. We use precision-recall curves and average precision (AP) per class to measure the performance of the algorithm.

**Evaluation:** The complete evaluation on VOC2007 test is given in Table 1. We compare our approach, named *DetectorNet*, to three related approaches. The first is a sliding window version of a DNN classifier by [14]. After training this network as a 21-way classifier (VOC classes and background), we generate bounding boxes with 8 different aspect ration and at 10 different scales paced 5 pixels apart. The smallest scale is $1/10$-th of the image size, while the largest covers the whole image. This results in approximately $150,000$ boxes per image. Each box is mapped affinely to the $225 \times 225$ receptive field. The detection score is computed by the softmax classifier. We reduce the number of the boxes by non-maximum suppression using Jaccard similarity of at least

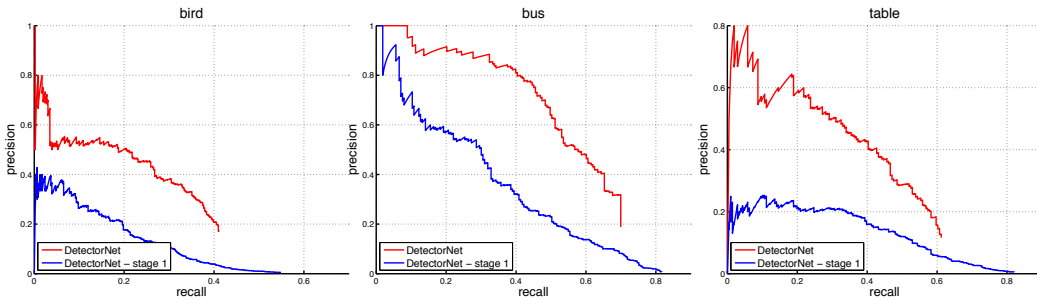

Figure 4: Precision recall curves of DetectorNet after the first stage and after the refinement.

0.5 to discard boxes. After the initial training, we performed two rounds of hard negative mining on the training set. This added two million examples to our original training set and has cut down the ratio of false positives.

The second approach is the 3-layer compositional model by [19] which can be considered a deep architecture. As a co-winner of VOC2011 this approach has shown excellent performance. Finally, we compare against the DPM by [9] and [11].

Although our comparison is somewhat unfair, as we trained on the larger VOC2012 training set, we show state-of-the art performance on most of the models: we outperform on 8 classes and perform on par on other 1. Note that it might be possible to tune the sliding window to perform on par with DetectorNet, however the sheer amount of network evaluations makes that approach infeasible while DetectorNet requires only ($\#$windows $\times$ $\#$mask types) $\sim 120$ crops per class to be evaluated. On a 12-core machine, our implementation took about 5-6 secs per image for each class.

Contrary to the widely cited DPM approach by [9], DetectorNet excels at deformable objects such as bird, cat, sheep, dog. This shows that it can handle less rigid objects in a better way while working well at the same time on rigid objects such as car, bus, etc.

We show examples of the detections in Fig. 3, where both the detected box as well as all five generated masks are visualized. It can be seen that the DetectorNet is capable of accurately finding not only large but also small objects. The generated masks are well localized and have almost no response outside the object. Such high-quality detector responses are hard to achieve and in this case are possible because of the expressive power of the DNN and its natural way of incorporating context.

The common misdetections are due to similarly looking objects (left object in last row of Fig. 3) or imprecise localization (right object in last row). The latter problem is due to the ambiguous definition of object extend by the training data – in some images only the head of the bird is visible while in others the full body. In many cases we might observe a detection of both the body and face if they are both present in the same image.

Finally, the refinement step contributes drastically to the quality of the detection. This can be seen in Fig. 4 where we show the precision vs recall of DetectorNet after the first stage of detection and after refinement. A noticeable improvement can be observed, mainly due to the fact that better localized true positives have their score boosted.

# 8   Conclusion

In this work we leverage the expressivity of DNNs for object detector. We show that the simple formulation of detection as DNN-base object mask regression can yield strong results when applied using a multi-scale course-to-fine procedure. These results come at some computational cost at training time – one needs to train a network per object type and mask type. As a future work we aim at reducing the cost by using a single network to detect objects of different classes and thus expand to a larger number of classes.

## Footnotes

[1]Trained on VOC2012 training and validation sets.

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
