[Reviews · NeurIPS 2013]

Submitted by Assigned_Reviewer_3

This paper proposes a deep neural network architecture for object detection. The core idea is to train the neural network to output segmentation masks, and then get bounding boxes out of these predicted masks.
Significance: As one of the first (in my knowledge) papers to use deep learning for PASCAL VOC-style object detection, this work is quite significant. The performance is comparable to state of the art object detection approaches, while the approach itself is considerably different from current object detection methods.

Originality: The paper is original to the best of my knowledge, both in terms of methods and in terms of results. The deep learning architecture draws from previous work, but the (crucial) step of going from a neural network to detections of multiple instances of objects is novel.

Quality: The paper is well evaluated. However, the authors compare to a much older version of the results of [9]. I would prefer a comparison to the newest DPM results, which can be obtained from http://www.cs.berkeley.edu/~rbg/latent/ . (This paper is better than the newest DPM release in 7 out of 20 categories, and worse or on par on others.)
A second concern I have is that the paper keeps the top 15 detections per image. This might discard high scoring detections and thus have a strong impact on the evaluation. It also seems to be a choice that is specific to this dataset. I would prefer if there was some stronger justification for the choice, and/or an evaluation without such a pruning.

Clarity: The paper is very clearly written. However, there are one or two details that should be there in the paper. The most notable being: the AP evqaluation requires the detector to output a score for each bounding box. What is the score here?
Summary: This paper is novel and significant, being one of the first papers to show convincing results on the hard detection problem using deep learning techniques. Provided the authors make the minor changes I mentioned above, this paper should be accepted.

Submitted by Assigned_Reviewer_6

This paper addresses the object detection problem using deep neural networks (DNNs). Recently DNNs have been successfully applied in image recognition and draw much attention from the vision community. This paper, however, moves a step forward to an arguably more challenging task: object detection. This work formulates detection as a regression problem, and uses DNNs to predict binary object masks directly from image pixels. A post-processing step of multi-scale refinement is used to further improve the detection accuracy.

The paper reports comparable results to the state-of-the-art deformable part models (DPMs) on the VOC 2007 benchmark. To my knowledge, this is the first paper that report the detection performance of DNNs on VOC dataset.

Pros:
1. A novel application of DNNs, and the paper gives an initial answer to an often discussed question: how DNNs work on object detection?
2. Comparable results to the existing state-of-the-arts.

Cons:
1.Some ad-hoc choices without clear justifications/evaluations, such as
- the choices of splitting the boxes into 4 halves. How did that affect the performance compared with using the full box only? Why is this splitting better than other arbitrary splittings (e.g. 3x3 grids), any insights?
- the second step refinement. Fig4 shows that the refinement dramatically improves the performance, but refinement step doesn't seem to be fundamental different to the first step of detection. I'm not clear where the boost is from. I don't see any new information is used. One explanation in the paper is that the refinement step provides higher resolution masks thus can detect smaller objects. Can't you just train a single step DNNs with proper parameters so that the resolution is high enough?
- The way of producing bounding boxes from the predicted binary mask sounds like a hack.
2. This work uses DNNs to make separate predictions on if each small cell is in an object or on the background. This is similar in spirit to the work of running classifiers on image pixels/superpixels to decide segmentations. (e.g. section 5 of "Semantic Texton Forests for Image Categorization and Segmentation" CVPR08). what do you think will happen if we replace the DNNs with other classifiers (e.g. random forest) on small image patches to generate the binary masks? It would be nice to have comparison with such a baseline.

A few questions/suggestions (These are not criticisms):
- I'd been curious to see more diagnosis on the error cases. Are they the same mistakes that DPMs make? (which I guess no).
- L343 says that you first trained the net for classification to start with good weights and later tuned the higher level layers. In the pre-training step, did you only use VOC2007 data, or extra data were used?
- The poor results of sliding window detector (second row in table 1) suggest that DNNs suffer from false positives. Any thoughts on improving its ability to prune the background patches?
- For reported results in Table 1, were there any filter/parameter sharing between classes?

Typos:
L21: extra "few"
L205: missing the closing parenthesis.
L429: "course-to-file" -> "coarse-to-fine"
Summary: The paper presents a novel applications of DNNs on object detection. This is the first paper that reports the detection performance on the VOC 2007 benchmark, which I think would be interesting to both learning and vision communities. However, the method itself is a bit hacky, I would recommend weak acceptance.

Submitted by Assigned_Reviewer_7

The authors propose a deep neural networks (DNN) for object detection. The DNN is developed on top of the model [14] designed for image classification. In stead of the softmax final layer as used in [14], the proposed networks use a regression model as the final layer. In order to overcome the detection challenges due to occlusion and small object size, a few more techniques are proposed: 1) multiple masks corresponding to lower, upper, left, right, and full object are generated; 2) a second stage DNN is applied on windows with multiple scale. During training, a large number of samples (10 millions per class) are sampled in order to train the complex DNN model.

Their final result on PASCAL 07 seems to be comparable to [9,19]. However, I do notice that the DPM performances reported on table.2 in http://www.cs.berkeley.edu/~rbg/latent/ are in many cases higher than the ones reported in table 1 in the paper. I also was surprised that sliding window based DNN performs so bad. I wonder if the model trained using the common bootstrapping (or hard negative mining) techniques, or not. Hence, I am not convinced that the regression-based method outperforms sliding window based method in a fair way.
Summary: This is a good attempt to improve object detection performance using the powerful DNN model. However, I am not convinced that the regression model + a few tricks is better than the sliding window based method. The method also has not yet demonstrated that DNN is the best method for object detection yet.
Author Feedback

Author rebuttal: We would like to thank the reviewers for their comments.

Reviewer 3:
1. We thank the reviewer for the pointer to the newest DPM results, which we will use to update the paper. In the submission we have used the published results as reported by the authors in their journal paper [9].
2. The decision to keep 15 bounding box from the first stage is based on a small subset of the training data in which we have estimated that 15 boxes cover all the objects of interest. The paper will be updated with this information.
3. We use the score of a DNN classifier trained on bounding boxes capturing examples of the 20 objects and background to score the final set of boxes as explained in lines 248 - 250. We will re-write these two lines to clearly state this. This is the score we use to plot PR curves and compute AP.

Reviewer 6:
1. In the design of the output of the DNN, we have tried to capture parts of the object in addition to the full object. Using 4 halves is in our opinion the simplest way to achieve this. Using a single full object mask instead of the five ones leads to worse results which we will report in Table 1 in the final submission.
2. The prediction of the final detection bounding boxes is inspired by the part-based models and formalized in a single cost function (see Eq.(3)). It combines, in our opinion, the five predicted masks in a clean and concise way.
3. The use of higher resolution output mask to avoid the refinement step, as suggested by the reviewer, is challenging because training a network to regress to a high-res mask would be computationally impossible for us.. As mentioned in lines 183 - 185, 24 x 24 = 576 output nodes result already in 2.4 million parameters in the last layer of the network alone. If we increase the resolution to 100 x 100 = 10000, for example, then the number of parameters will go to 40 million just in this last layer which makes it prohibitive to train within our resource constraints. Hence we opted for coarse-to-fine procedure where in the refinement step we apply the net on a subwindow resulting in a higher resolution output w.r.t the full image. We will add this more detailed description to after the lines 185 for better clarification.
3. The main difference between ours and Semantic Texton Forests (STFs) is that DNNs derive their prediction for each cell from the full image while STFs work on patches to classify each segment. Thus we believe that DNNs are capable of naturally expressing global interactions and contextual information not present in the STF setup.
4. The main difference between DNNs and DPM is that DNN work better with deformable object such as most of the animals. It seems that the network is capable of learning representation nicely capturing such deformations. On the other side, DPMs localize better small rigid objects. This is mainly due to the fact that the net has a limit on its output resolution.
5. For initialization we used a network trained on the 21-way classification problem of classifying the cropped 20 object classes and background from VOC2007 training (the same net used for scoring of our output as well as the sliding window experiment). We used the parameters of all the layers except the last one.
6. Perhaps the use of more background patches would help to improve the sliding window DNN.
7. We use the same hyperparameters for all classes in Table1.

Reviewer 7:
1. We have trained the DNN classifier, which we apply for the sliding window evaluation, using a large number of bounding boxes containing background in addition to boxes containing examples of the 20 classes of interest. We use 10 times more background boxes than examples of an indivisual class. In order to improve localization, the training set also contained a very large number of crops that partially overlap the objects with a threshold less than 0.4. The resulting model has a very narrow response range: giving high confidence only to fairly covered objects. The model is trained to be high quality: we use it to score our bounding box prediction after the second refinement step.
2. The formulation of the detection problem as regression is quite different in spirit from the huge body of detection work, based mainly on part-based models or segmentation, and therefore we believe it is quite novel and not incremental.
3. The use of a coarse-to-fine refinement and non-max suppression are widely used techniques in computer vision which have been traditionally widely used.